# Platelet activation near point-like source of agonist: Experimental insights and computational model

**Ezhena S. Starodubtseva[1], Tatyana Yu. Karogodina[1,2], Alexander E. Moskalensky [1] ***

1 Laboratory of Optics and Dynamics of Biological Systems, Novosibirsk State University, Novosibirsk, Russia, 2 Laboratory of Photoactivatable Processes, N.N. Vorozhtsov Novosibirsk Institute of Organic Chemistry SB RAS, Novosibirsk, Russia

\* sunmosk@mail.ru

**Data Availability Statement:** All relevant data are within the manuscript and its Supporting Information files

## Abstract

Disorders of hemostasis resulting in bleeding or thrombosis are leading cause of mortality in the world. Blood platelets are main players in hemostasis, providing the primary response to the vessel wall injury. In this case, they rapidly switch to the activated state in reaction to the exposed chemical substances such as ADP, collagen and thrombin. Molecular mechanisms of platelet activation are known, and detailed computational models are available. However, they are too complicated for large-scale problems (e.g. simulation of the thrombus growth) where less detailed models are required, which still should take into account the variation of agonist concentration and heterogeneity of platelets. In this paper, we present a simple model of the platelet population response to a spatially inhomogeneous stimulus. First, computational nodes modeling platelets are placed randomly in space. Each platelet is assigned the specific threshold for agonist, which determines whether it becomes activated at a given time. The distribution of the threshold value in a population is assumed to be log-normal. The model was validated against experimental data in a specially designed system, where the photorelease of ADP was caused by localized laser stimulus. In this system, a concentration of ADP obeys 2-dimensional Gaussian distribution which broadens due to the diffusion. The response of platelets to the point-like source of ADP is successfully described by the presented model. Our results advance the understanding of platelet function during hemostatic response. The simulation approach can be incorporated into larger computational models of thrombus formation.

## Introduction

Platelets are known to be involved in many physiological processes: apart from their main hemostatic function, they also participate in immune and inflammatory responses, release growth factors and regulate angiogenesis [1–3]. However, the primary function of a platelet is to prevent bleeding in case of vessel wall injury. Platelets have developed a unique activation mechanism which enables fast transformation from resting, inert cells to particles eager to

**Funding:** The study was supported by the Russian Science Foundation (grant #23-75-10049). The funders had no role in study design, data collection and analysis, decision to publish, or preparation of the manuscript.

**Competing interests:** NO authors have competing interests

adhere to the damaged surface and aggregate with each other [4]. Apart from adhesion and aggregation, platelets also may release granules recruiting additional cells and provide factors promoting blood coagulation [5].

Biochemical cascade of the platelet activation involves special chemical compounds called agonists, which accumulate around the damaged area and trigger the process [6]. The agonists, such as collagen, adenosine diphosphate (ADP), epinephrine, thrombin etc., bind to the specific receptors on the platelet plasma membrane and initiate a physiological response [7, 8]. Namely, ADP induces platelet activation through the interactions with G-Protein-Coupled Receptors (GPCRs) P2Y$_1$ and P2Y$_{12}$ [9]. Both of them have an impact on phospholipase C (PLC) that leads to the release of inositol trisphosphate (IP3), which, in turn, activate fast elevation of Ca$^{2+}$ in the cytoplasm of the cells [10, 11]. The rise in cytoplasmic calcium concentration is considered to be the hallmark of platelet activation.

The dynamics of cytosolic calcium concentration can be detected in real time using the fluorogenic calcium probes such as Fluo-4 [12]. Calcium spikes can even be visualized in single platelets with the fluorescent microscopy [13], which allows one to evaluate the activation of each cell. This method is suitable for investigating activation by an agonist with spatially inhomogeneous distribution, which is the case in physiological processes. Dynamic inhomogeneous distribution may arise from the diffusion of agonists from localized sources. It is of great interest since the hemostatic plug is formed by the aggregation of the activated platelets due to the diffusion of thrombin and ADP from its core [14]. In addition, granules released by activated platelets that contain additional activators acting as positive-feedback amplifiers [15, 16] also contribute to the agonist concentration profile. However, granule release usually requires high concentrations of ADP [17] and occurs in a later stage of activation [18]. Nevertheless, in order to model the clot formation, it is important to take into the account spatiotemporal distribution of the agonists; however, direct experimental observations of platelet activation in a system with inhomogeneous distribution of ADP are still lacking.

Molecular mechanisms of platelet activation are known. For instance, Purvis et al. [19] developed a detailed model which is able to predict cytosolic calcium concentration in response to the addition of ADP. Although the response of the platelet population has monotonic dependence on the agonist concentration, activation of each platelet has stochastic, threshold-like nature. Sveshnikova et al. [20] identified the decision-making element determining the activation threshold. Modeling of thrombus formation implies inhomogeneous distribution of agonists, and the activation state of each platelet should be determined [21, 22]. Additionally, both agonists and cells are non-stationary due to blood flow and diffusion [23–25]. Usually, simpler models of platelet activation are used in such complex problems [26–28]. Hill equation was used to describe threshold-like dependence of activation probability on the ADP concentration [29, 30]. Litvinenko et al. [31] assumed that the sensitivity of platelets to ADP has log-normal distribution. These simplified models enable complex simulations, but there is an urgent need for their experimental verification.

In this study, we present a simple computational model of the platelet population response to a spatially inhomogeneous stimulus. First, computational nodes modeling one or few platelets are placed randomly in space. Each node is assigned the specific threshold for agonist, which determines whether it becomes activated. The distribution of the threshold value in a population is assumed to be log-normal. The model is validated against experimental data in a specially designed 2-dimensional system. The platelets are activated by optically uncaging of ADP from its photolabile analogue, NPE-caged ADP. Since the photorelease of ADP is caused by localized laser stimulus, the concentration of ADP obeys 2-dimensional Gaussian distribution which broadens due to the diffusion. The developed algorithm allows us to simulate

platelets population response (fraction of activated cells and average fluorescent signal) to such an inhomogeneous stimulus.

## Methods

### Sample preparation

The blood samples of three healthy donors (female 22yo, male 35yo and female 40yo) were obtained from the cubital vein. Written informed consent was obtained from the volunteers prior to the study. The study protocol was approved by the Ethics Committee of the research Institute of Clinical and Experimental Lymphology–Branch of the Institute of Cytology and Genetics, Siberian Branch of Russian Academy of Sciences. The volunteers were recruited during the period from 01/09/2023 to 01/04/2024.

The samples were collected to the standard vacuum tubes containing sodium citrate as anticoagulant (9:1). After that blood sample was settled at room temperature for about one hour until a layer of platelet-rich plasma was separated from the rest of the blood cells. Centrifugation was not used. Then fluorescent calcium probe Fluo-4-AM (Thermo Fisher Scientific, USA) was added to the sample for labeling. The 1 μL of Fluo-4-AM (1 mM stock solution in DMSO) was added to 61.5 μL to phosphate buffered saline (PBS) and mixed with blood plasma in 1 to 1 proportion. After 30 minute incubation, 8 μL of the labeled cells were placed in wells of 8-well polystyrene strip plate with addition of 3 μL of caged ADP (NPE-caged-ADP, 10 mM stock solution) and 69 μL of PBS (final concentration of caged ADP is 0.375 mM). We tested out HEPES buffer instead of PBS in additional experiment, but results were controversial (S3 Fig in S1 File). After another 30 minutes the samples were used for the experiment. The height of the liquid in each sample was 2 mm.

### Experimental system

For the observation of changes in platelets' fluorescent signals during activation inverted microscope Carl Zeiss AxioVert.A1 was used. It is reported that the maximum of the fluorescence of Fluo-4 bound with $Ca^{2+}$ is at the wavelength of 516 nm when it is excited with the light of a wavelength of 494 nm which corresponds with the wavelength of LED in microscope. For the purpose of detection and recording the platelet activation, a high-sensitive Axiocam 503 mono camera was used. 10X dry objective was used to observe many cells.

To locally induce the platelet activation we used a laser DTL-375QT with a wavelength of 355 nm. The laser beam was redirected into the well with a sample by mirror (Edmindoptics, #33–504) and focused with the lens (Thorlabs, AC254-045-A-ML) with a focal length of 45 mm, the observable size of the laser spot was around 80 μm. In addition, an optical filter was added to the system to cut off the secondary harmonic of the laser at 532 nm which otherwise resulted in noise in fluorescence signal. The control of the duration of laser flash and the time of its appearance was carried out through the usage of an Arduino Nano board that was connected through to the PC and the laser control unit. The board acts as impulse generator with the frequency of 2 kHz, which triggers laser pulses. The duration of the pulse tray is controlled by a PC operator. In our experiments, the duration was 200 ms, which was sufficient for the uncaging of ADP and platelet activation.

As a result, the following experiment was performed (Fig 1). Samples were placed on the scanning table of the microscope and rested for 5 minutes until the cells settled. The recording of the video with a frame rate of 3 fps began and approximately a minute after laser flash provokes photodissociation of caged ADP so the released ADP diffuses and triggers the platelet activation. The duration of the received video was around 3 to 4 minutes.

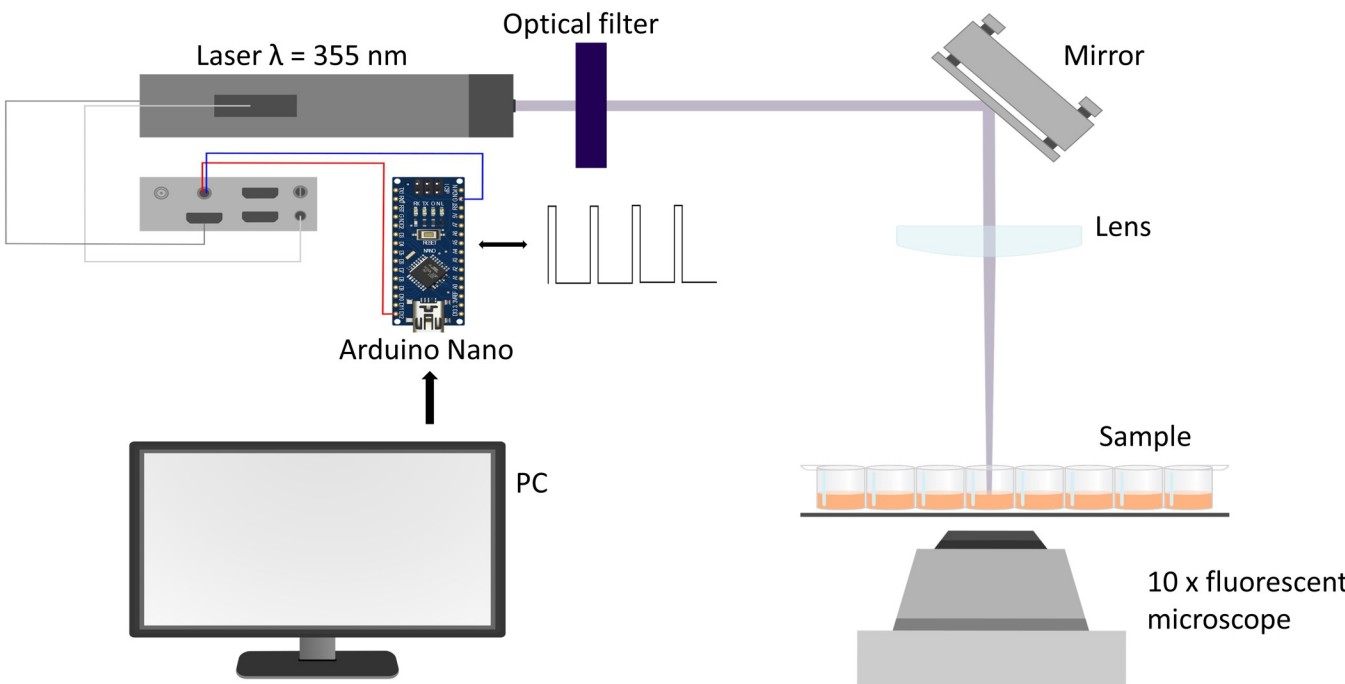

**Fig 1. Scheme of the experimental setup.** Pulsed laser with the wavelength of 355 nm is directed to the experimental sample containing platelets with photolabile caged ADP. The laser is enabled for a computer-controlled duration, inducing release of active ADP and platelet activation, which is monitored with fluorescence microscope using Fluo-4 calcium probe.

## Treatment of experimental data

Analysis of the recorded video was performed using the ImageJ (FiJi) software. To calculate the mean fluorescence signal *versus* time in the areas of interest, we used standard procedure utilizing ROI manager and the multi-measure feature. For the analysis of single cells, Track-Mate plugin was used [32]. It allows one to automatically detect cells and provides the average intensities over time even if the cells are moving. The obtained results (coordinates and mean fluorescence intensity for each platelet) was saved and used in a custom Python script, which was needed for the determination of calcium spikes. Calcium spikes were determined using the standard peak detection function from the SciPy library. As a result, we were able to obtain the time of first calcium spike as a function of coordinates, which is needed for the validation of theoretical model.

## Mathematical model of the platelet population response to agonist

The interaction between the platelets and agonists results in the activation response. This process can be considered as a switch from resting to activated state when the concentration of agonist exceeds certain threshold. The state switch triggers a cascade of biochemical reactions leading to the changes in the intracellular calcium concentration. To simulate the cellular response, we developed an algorithm that allows to predict the fraction of activated platelets over time in complex environment with inhomogeneous distribution of agonist.

The outline of the algorithm is shown in the Fig 2A for the case of two spatial dimensions. It is based on the assumption that all platelets are uniformly distributed on the area of interest and the sensitivity threshold TR belongs to the log-normal distribution:

$$\text{TR} \in \text{logN}(\text{TR}; \mu, \sigma^2), \tag{1}$$

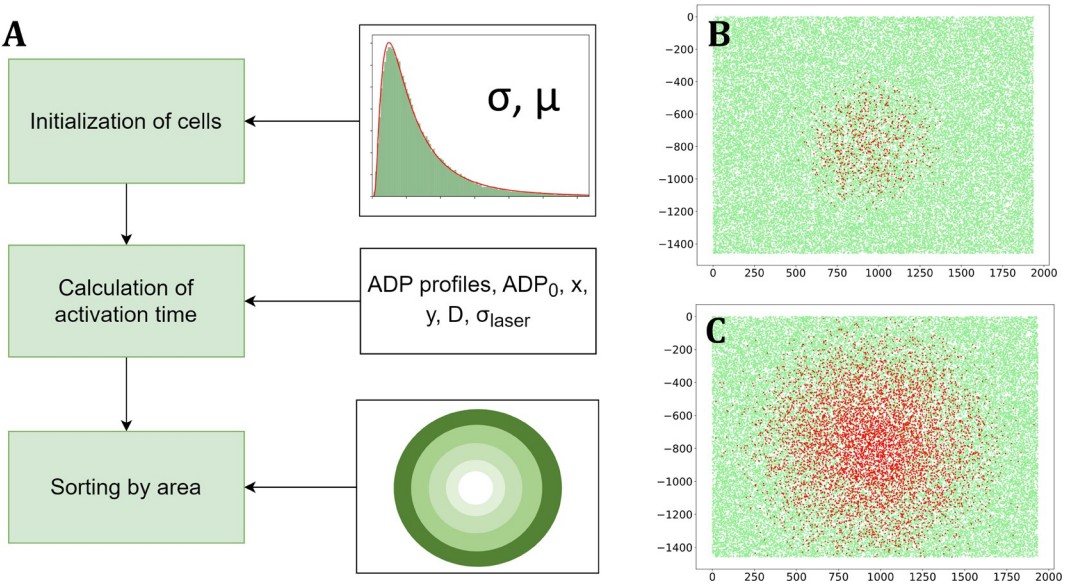

**Fig 2. Simulation of diffusion-derived platelet activation.** A. Scheme of the algorithm. B. Simulated platelets just after the localized activation stimulus: each dot represents single cell, the red ones are activated platelets and the green ones are non-activated. C. The same after 50 sec of the stimulus onset, showing the diffusion-mediated spread of the area of activated cells.

where

$$\mathrm{LogN}(\mu, \sigma^2) = \frac{1}{\sqrt{2\pi}\sigma x} \exp\left[-\frac{(lnx - \mu)^2}{2\sigma^2}\right] \tag{2}$$

and $x$ is the threshold value.

The computation starts with initialization of cells. Typically, thousands of platelets are randomly placed in the area of interest, each having their own coordinates $(x_i, y_i)$ and the ADP sensitivity threshold $TR_i$. We presume that the displacement of platelets is insignificant and their size is small compared to characteristic scale of ADP concentration change; therefore, all the objects are considered immobile and point-like.

Next, the algorithm takes the ADP concentration profile $ADP(x, y, t)$ as an input and compares the local ADP value for each platelet with the threshold $TR_i$. The comparison is made sequentially in time. Initially, all platelets are considered to be in resting state and they become activated when the condition $ADP(x_i, y_i, t) > TR_i$ is met for the first time. Thus, the algorithm determines the moment of activation for each simulated platelet, which we designate as $TA_i$.

Fig 2B, 2C show the example of the graphical representation of the results: each dot represents a platelet; the red ones correspond to the activated cells and the green ones are non-activated. All cells can be sorted by area, for instance, to determine the fraction of activated cells in the specific area of interest.

In this paper, we compare the theoretical prediction with experimental measurements of platelets intracellular calcium. Previous experiments showed that there is a delay (lag time) between the trigger pulse and the first calcium spike, which is distributed log-normally [33]. Therefore, to calculate the time of the first calcium spike $TS_i$, additional delay $TD_i$ should be added to the activation time:

$$TS_i = TA_i + TD_i, \tag{3}$$

where $TD_i$ belongs to the log-normal distribution with parameters $\mu_D, \sigma_D$:

$$\text{TD} \in \text{LogN}\left(\mu_D, \sigma_D^2\right). \tag{4}$$

## ADP diffusion model

In this study, ADP is used to trigger platelet activation. The agonist is released from its caged analogue by the laser pulse. Since the laser beam has Gaussian intensity profile, we decided that the equations that describe the diffusion of a two-dimensional Gaussian peak [34] can be applied for the ADP concentration profiles:

$$[\text{ADP}] = [\text{ADP}_0]\frac{t_0}{t + t_0}\exp\left[-\frac{r^2}{4D(t + t_0)}\right] \tag{5}$$

where $\text{ADP}_0$ is the initial concentration of a released agonist in the sample, $D$ is the diffusion coefficient, $t$ is the time since the system's excitation, $r$ is the distance from the excitation center and $t_0$ is the characteristic time defined from the Gaussian distribution so that the distribution width at $t = 0$ matches the width of the laser beam.

## Results

The following procedure was used to estimate the effect of ADP diffusion on the platelet activation. Firstly, platelets were prepared as described in Methods section. Secondly, the samples were placed on the experimental set up (Fig 1) and the videos were recorded. The examples of received frames are shown in Fig 3A–3D. Each platelet is represented as a glowing dot due to a fluorescence of the calcium probe showing a baseline level of intracellular calcium. After recording of ~60s of the resting sample, a laser flash was applied to the central region of the frame as showed in Fig 3B. This triggers a photorelease of ADP from its "caged" analogue inside the circle. The figure also shows the concentric circular areas with widening diameter which were used for later analysis to measure mean fluorescent signals over time. The diameter of the central circle is 108 μm, while the thickness of the other areas is 36 μm. Mean fluorescence signal rapidly increased near the irradiation circle due to platelet activation, as shown in Fig 3C. The area in which platelets are activated expands due to the diffusion of ADP; Fig 3C shows the frame after 10 s after irradiation, whereas Fig 3D shows the frame after 40s.

Fig 3E–3J represent typical single-cell calcium signaling in representative platelets. Rapid spikes with further decrease indicate the activation response, as observed in classical studies [13, 35]. Some platelets were spontaneously activated before the ADP uncaging, but the majority of cells were stable before the laser irradiation. The general tendency is clearly visible: platelets from the area nearest to the irradiation spot activate much faster than that from more distant areas. In Fig 3E the calcium spikes occur several seconds after the laser flash, while in subsequent plots the shift becomes larger: in Fig 3J the response delay is more than 20 s.

Fig 4A shows how the ADP concentration profiles change over time due to the diffusion according to Eq. (5). The initial concentration of the released ADP was estimated with the high-performance liquid chromatography (HPLC) to be $\text{ADP}_0 = 1.5 \cdot 10^{-5}$ M, whereas the initial width of the Gaussian corresponds to the width of the laser beam which was determined from video.

The concentration of ADP at the release point is much higher than at the periphery of the frame, but over time the ADP level in this area drops while increasing at distant points, which explains the delayed activation of cells. To quantify the activation time, we performed analysis of single-cell calcium response, looking for the first calcium spike after the laser flash. Fig 4C shows the distribution of the activation time (more precisely, the time of appearance of the

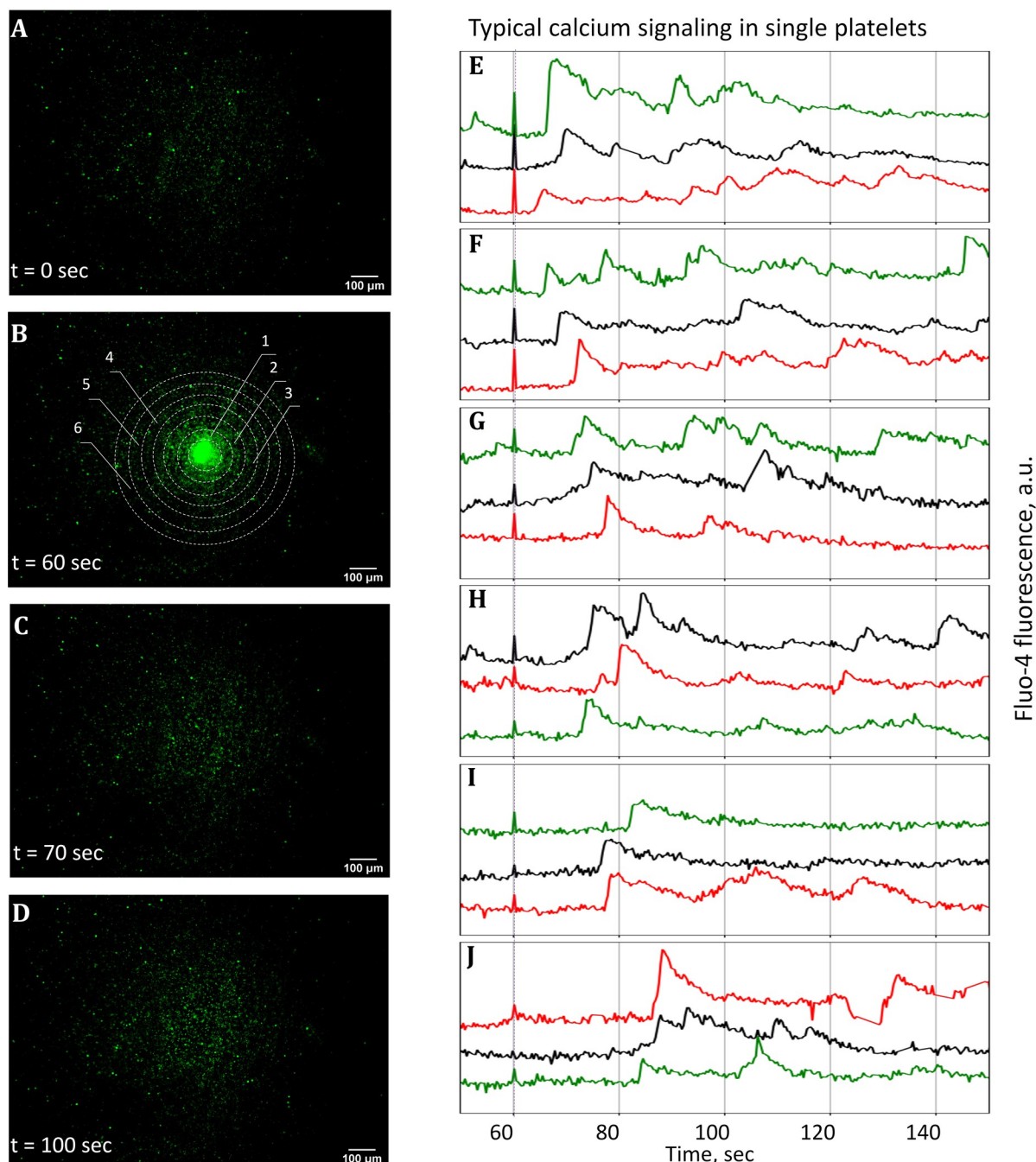

**Fig 3. Dynamics of activation spreading.** A. Resting platelets before the stimulus. B. Initiation of ADP's photorelease via the laser irradiation. Enumerated circles outline the regions in which fluorescent signals were measured. C. 10 seconds after the appearance of laser flash. Increasing in fluorescence levels of platelets that were nearby exposed area are visible whereas there are no changes in distant regions. D. 40 seconds after the appearance of laser flash. Spreading of activation due to the diffusion of ADP is visible. E-J. Calcium dynamics in representative cells from corresponding marked areas (areas 1 to 6 in panel B, respectively). Dashed line across all graphs represents the time when laser flash was applied. The exposure time is 1/3 s.

first calcium spike TS, corresponding to Eq (3)) for four separate concentric areas. The appearance of platelets with delayed activation is visible as widening of the distribution to the right.

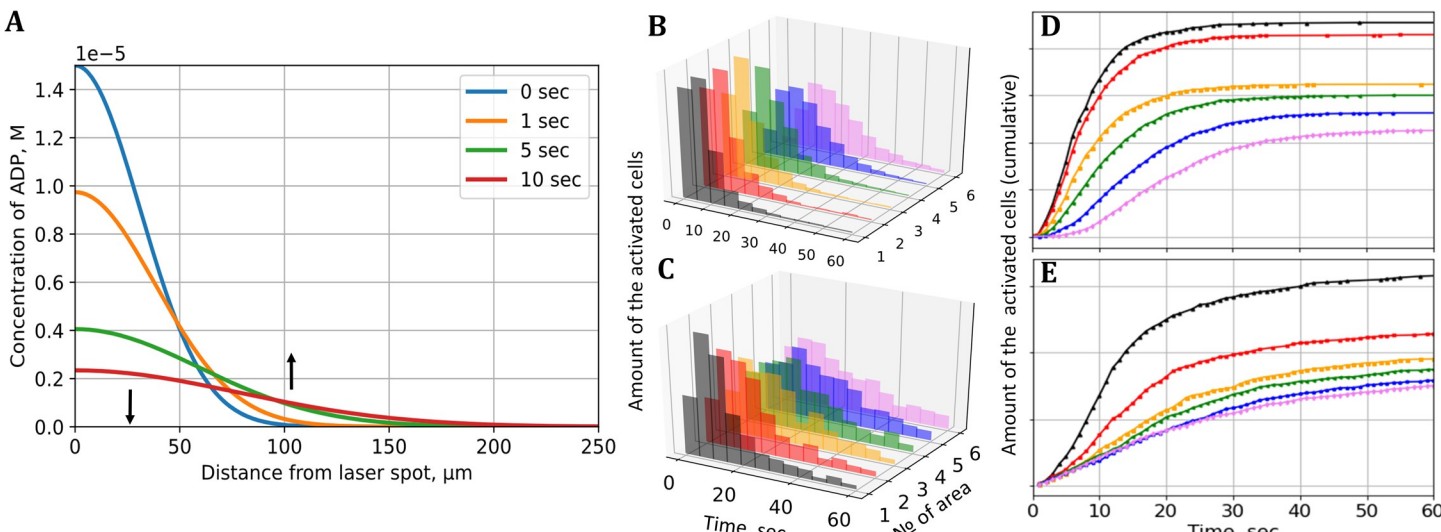

**Fig 4. Spatiotemporal ADP concentration profiles.** A. Calculated ADP profile changes over time. B. The histogram of the activation time obtained in simulation. C. The histograms of the first spike time calculated for platelets in the corresponding areas. D. The fraction of activated platelets in simulation. E. The fraction of activated platelets in the experiment.

The mathematical model described in Methods, Eqs (1)-(4), allows us to simulate the platelet response to agonist given the time-dependent ADP concentration profile. To compare this model with experimental data, we used the ADP diffusion coefficient D = 257 μm²/s from [36].We assume that the activation time delay TA is negligible for the area nearest to the irradiation circle, which can be estimated according to the model and characteristic parameters of platelets; therefore, we were able to determine the parameters of the delay time TD by fitting of experimental histogram for this circle with log-normal distribution density similar to [33]. The obtained parameters are given in S1 Table in S1 File. The simulation results for N = 100000 platelets are shown in Fig 4B. Widening of the distributions for distant areas are also visible, which indicates the qualitative agreement with experimental findings. The fraction of activated platelets as a function of time is plotted in Fig 4D and 4E for theory and experiment, respectively. Although the overall results are in qualitative agreement, simulation predicts the lag-time in distant areas, which is absent in experiment.

Next, we applied our model to the analysis of mean fluorescence signals from the separate areas. In this case, we do not need to use single-cell tracking and analysis software but simply use the built-in "multi-measure" function of ImageJ. The primary processing of the experimental data includes few steps. Firstly, the initial data set is divided into 2 parts, before and after the laser flash. During the first part system is stabilizing and the rapid rise in the signal is observed after the laser flash. The data for each area was normalized to the intensity of initial signals (before the flash). Fig 5A, 5C and 5E shows the obtained data for three experiments (for each of three donors) with duration of irradiation of 200 ms and the concentration of the caged ADP of 0.375 mM. The signal growth is faster near the irradiation circle (black line) and slower at distant areas, displaying the diffusion-mediated delay.

To simulate the fluorescence signals, we use only Eqs (1)-(2) because we do not need to analyze individual calcium spikes. Instead, we made the following simplification. While in experiment the mean signal is comprised by many calcium spikes which occurs in different cells at different times, in a simulation we consider each cell to contribute a smooth, growth-and-decay function to a total signal. This is justified because there are many cells in the areas under

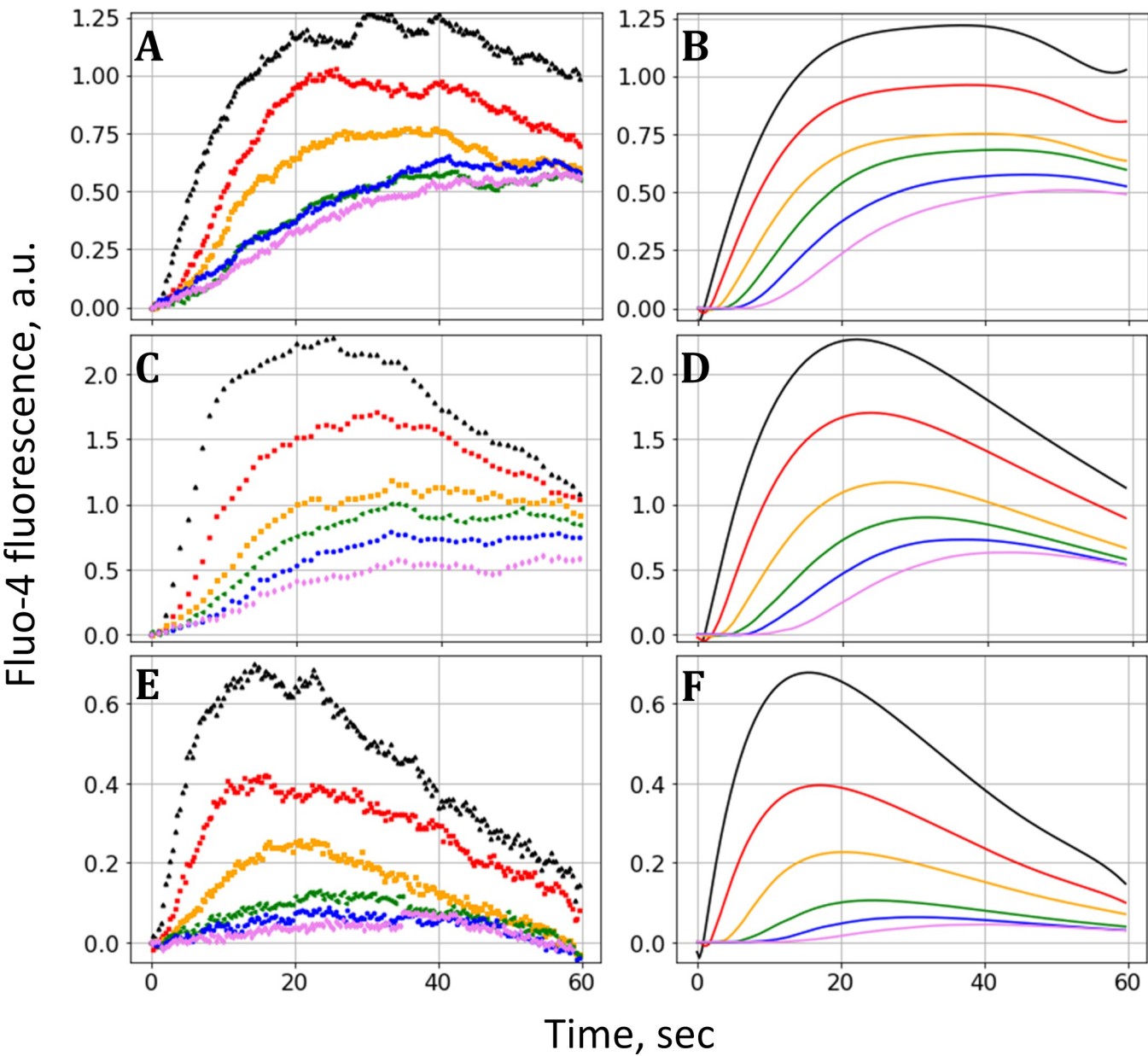

**Fig 5. Simulation of fluorescent signals.** A,C,E: Mean fluorescence intensity in the concentric circles around the laser beam (black curve–nearest circle, pink–the most distant circle) shown for three donors. B,D,F: The best-fit theoretical curves corresponding to the curves on the left part of the figure.

consideration and all spikes are well averaged; in other words, each "cell" in the simulation can be considered as an ensemble of many cells. The function itself is determined from experiment again using the area nearest to the irradiation circle, where the delay is considered to be negligible. We constructed a basic fluorescence intensity function by polynomial fit of the corresponding experimental time-dependence. Then the fluorescent signals in other areas are determined by the summation of the basic function shifted in time by $TA_i$ for all cells. Lastly, the obtained intensities are normalized by the number of cells simulated in the area.

Calculations were performed for $N = 50000$ simulated objects. In this case, we varied σ and μ to determine the best-fit parameters by a least square method (LSM) resulting in

minimal discrepancy between experimental and simulated data. This approach could be useful for the estimation of platelets population sensitivity, which may have the diagnostic value. The obtained parameters for three donors are presented in the Table 1.

The obtained sensitivity parameters, namely the mode of distribution that is near $10^{-7}$ M, are in range of the previously published values, in particular, the platelet sensitivity parameter MPS-ADP which was determined in [31] by measuring the shape change. As it is shown on the best-fit approximation graphs (Fig 5B, 5D and 5F), the model predicts characteristic delays in distant areas as well as the decrease in fluorescence growth rate.

## Discussion

In this paper, we present novel simplified model of platelet population spatiotemporal response on known agonist, ADP. The developed model is used for prediction of cellular threshold-like activation behavior on the surrounding microenvironment. It combines calculations of reaction time according to the spatiotemporal distribution of ADP and fluorescent signals for each simulated cell. Results showed qualitative correspondence of simulation with experimental data in the experiment of locally induced platelet activation. We assume that suggested approach can be utilized in large-scale models of thrombus growth. Furthermore, the received parameters of log-normal distribution that indicate the sensitivity of platelet population can be valuable as a diagnostic tool.

The described approach definitely has some limitations. First, the model is phenomenological and extremely simplified. It does not include any kinetic equations that describe biochemical machinery of signaling pathways, making it impossible to predict the shift in the model parameters and even its validity due upon the change in real-world variables. Second, as it was described earlier, we used the continuous smooth polynomial function for the simulation of calcium response for an ensemble of cells, which is inapplicable if the ensemble size is small because the calcium signaling in single cells consists of many stochastic spikes. This response is quite complicated because of unpredictability of spacing between calcium spikes and their amplitudes. As a result, we have to use experimental data as a model input–for instance, we used the histogram of response time TD in the single-cell experiments and mean calcium signal of platelets near the ADP source as a basic function for the description of cells' ensemble response. On the other hand, this gives our model the ability to be used for the description of any experiments, not necessarily calcium response. The study of effects other than the calcium signaling might shed light in different aspects of platelet function. For instance, the activation of surface integrins, granule release and procoagulant surface formation can be visualized using special fluorescent labels [37]. The platelet shape change [38, 39] is also a sensitive marker of platelet activation. All these effects are also affected by the inhomogeneous distribution of agonist and follow the time-shift pattern of activation, which is predicted by our model.

Some divergence between the model and experimental data is caused by deliberate exclusion of some issues. First, we do not take into the account the positive feedback loop caused by the granule release [40] and crosstalk between signaling pathways. Although it is possible to estimate granule release in living cells [41], further interactions of granules with platelets as well as their impact on activation compared to the processes described in this paper are hard to predict. Second, we assume the presence of only activated and non-activated platelets in the

**Table 1. The parameters of the platelet ADP threshold distribution for three healthy donors.**

| Donor, № | $\mu$ | $\sigma$ | ADP threshold, median (M) | ADP threshold, mode (M) |
|---|---|---|---|---|
| 1 (Fig 5A, 5B) | -15.5 | 2.7 | $1.83 \cdot 10^{-7}$ | $1.25 \cdot 10^{-10}$ |
| 2 (Fig 5C, 5D) | -13.2 | 2.7 | $1.88 \cdot 10^{-6}$ | $1.43 \cdot 10^{-9}$ |
| 3 (Fig 5E, 5F) | -14.5 | 1.7 | $4.99 \cdot 10^{-7}$ | $2.68 \cdot 10^{-8}$ |

sample. However, there also other types of cells such as partially activated platelets [42], aggregating platelets that are a strong source of molecules that boost procoagulant activity and procoagulant platelets which release inflammation mediators [43]. Lastly, the developed model for simplicity suggests identical activation response for all cells, while there should be a correlation between platelet sensitivity and the response speed and amplitude.

Further researches might cover the above-mentioned issues alongside with the determination of synergetic and inhibitory effects of multiple agonists. Our recent studies [44, 45], showed that the use of photo-generated nitric oxide (NO) inhibits spontaneous platelet activation, providing more physiological conditions *in vitro*. It is interesting to combine ADP with other photolabile compounds affecting platelet activation, such as caged epinephrine, arachidonic acid [46] and others, especially if they can be "uncaged" independently at different wavelengths. Further studies might broaden the understanding of platelet functions and help to find out the possible ways of treatment of cardiovascular diseases.

## Supporting information

**S1 File.**
(PDF)

**S1 Video.**
(MP4)

**S1 Data.**
(ZIP)

**S2 Data.**
(GIF)

## Author Contributions

**Conceptualization:** Alexander E. Moskalensky.

**Data curation:** Ezhena S. Starodubtseva.

**Formal analysis:** Ezhena S. Starodubtseva.

**Funding acquisition:** Alexander E. Moskalensky.

**Investigation:** Ezhena S. Starodubtseva, Tatyana Yu. Karogodina.

**Software:** Ezhena S. Starodubtseva.

**Supervision:** Alexander E. Moskalensky.

**Visualization:** Ezhena S. Starodubtseva.

**Writing – original draft:** Ezhena S. Starodubtseva.

**Writing – review & editing:** Alexander E. Moskalensky.

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
