## [Decision Letter · Decision Letter 0]

7 Jun 2024

PONE-D-24-12691Platelet activation near point-like source of agonist: theoretical model and experimental verificationPLOS ONE

Dear Dr. Moskalensky,

Thank you for submitting your manuscript to PLOS ONE. After careful consideration, we feel that it has merit but does not fully meet PLOS ONE’s publication criteria as it currently stands. Therefore, we invite you to submit a revised version of the manuscript that addresses the points raised during the review process. The reviewers have provided incisive and valuable feedback that needs to be addressed in its entirety for publication in PLOS ONE. Please note that both reviewers took exception to the statement in the abstract that “...existing models are too complicated for large-scale problems, for instance, simulation of the thrombus growth, where the concentration of agonist varies significantly and heterogeneity of platelets should be taken into account.” Please ensure that in addition to addressing the remaining comments, that you address this statement and cite publications that solve for thrombus growth with agonist gradients and individual platelets.

We look forward to receiving your revised manuscript.

Kind regards,

Heather Faith Pidcoke, MD, MSCI, PhD

Academic Editor

PLOS ONE

Journal Requirements:

"The study was supported by the Russian Science Foundation (grant #23-75-10049)."

Reviewers' comments:

Reviewer's Responses to Questions

**Comments to the Author**

1. Is the manuscript technically sound, and do the data support the conclusions?

Reviewer #1: Yes

Reviewer #2: Yes

2. Has the statistical analysis been performed appropriately and rigorously? 

Reviewer #1: Yes

Reviewer #2: N/A

3. Have the authors made all data underlying the findings in their manuscript fully available?

Reviewer #1: Yes

Reviewer #2: Yes

4. Is the manuscript presented in an intelligible fashion and written in standard English?

Reviewer #1: Yes

Reviewer #2: Yes

5. Review Comments to the Author

Reviewer #1: Authors study a novel system of photo-released ADP and platelet activation in space and time. The experiments and data analysis are conducted with care.

Major comments.

The paper does not appear to include conditions where agonist induced ADP release and thromboxane synthesis play an autocrinic role. Authors are encouraged to find conditions of platelet density, P2Y12 inhibitors, or COX1 inhibitors that may modulate the spread of platelet activation. Are there conditions where platelet release of ADP causes a self-sustaining wave of calcium activation, distinct from the diffusional spread of photo-released ADP. As a thrombus grows, the platelet density is very high and autocrine activation definitely occurs.

The authors should discuss the role of geometry and platelet density. The height of the liquid determines the size of the reaction volume and the amount of photo-released ADP. Different results would be obtained in different geometries such as activation of a cross section of a tubular geometry or rectangular geometry (as in microfluidics).

Phosphate buffer will cause calcium to precipitate. The studies should be repeated with HEPES buffered saline. (Tris should be avoided since it has anti-protease activity in coagulation studies).

The authors should provide evidence that the platelets have settled to the bottom of the well. If the platelets have spread then they are partially activated before the laser is used.

Minor Comments:

Abstract: “However, existing models are too complicated for

large-scale problems, for instance, simulation of the thrombus growth, where the

concentration of agonist varies significantly and heterogeneity of platelets should be

taken into account.” This statement is not correct. There are several publications that solve for thrombus growth with agonist gradients and individual platelets.

Reviewer #2: The manuscript reports a study of platelet activation by a point source of ADP. The Authors designed an experimental setup and developed a mathematical model for spatio-temporal dynamics of the diffusion-mediated spread of the area of activated cells. The Materials and Methods are detailed enough to allow the reproducibility of the results. The Authors replied all the remarks from the previous round of peer-review, even added new experiments with healthy donors, as was required by the Reviewer #1.

1. The paper in written in standard English, however, some typos are present, e.g., on page 2, line 13 - "Biochemical cascade of the platelet activation involve special" - should be "involves".

2. In the PDF version of the manuscript some equations (e.g. 1 and 2) are not displayed properly, although in the docx-file the equations are readable. This should be elaborated during the publication process in case of acceptance.

3. The Authors "presumed that the displacement of the platelets as well as their size is insignificant". I can agree that this assumption is somehow valid in terms of the displacement. However, the independence of the activation threshold (minimal ADP concentration) from platelet size should be substantiated, as it is not obvious. Could there be a correlation between the ADP-threshold and platelet size? Some references suggest that platelet volume may be a risk marker for platelet activation:

https://doi.org/10.1080/095371002220148332

https://doi.org/10.1093/rheumatology/kew437

4. Fig.3. In the caption it should be explicitly mentioned that panels E-J correspond to the areas 1-6 marked on the panel B.

5. The Authors claim that existing mathematical approaches are too complicated to be used in systems biology models. But in my opinion their model oversimplifies the picture. Generally, the paper tells us that the diffusion of ADP activates the platelets. The model is based on the solution of the diffusion equation (Fick's law) and the probabilistic distribution of platelet susceptibility to ADP. This finding does not reveal any new biophysics mechanism, nor it incorporates already known features into the suggested model, such as granule release and positive feedback loops. In my opinion, the paper is valuable for its experimental part (as the results seem rigorous), while the proposed model is essentially phenomenological and it doesn't enhance our understanding of platelet activation.

Therefore, I suggest to modify the title: the experimental part should brought forward, while the theoretical model is subsidiary. The fact that the theory "captures main features of experiment (activation spread) but cannot accurately describe other features" mainly says that the model is not precise enough, as it doesn't account for possible nonlinear effects and feed-back loops, which are essential for such an active medium. Clearly, the spreading of "platelet activation" in not entirely governed by the diffusion, although the diffusion of ADP indeed triggers the observed processes.

I also suggest to remove the phrase "detailed models require too much computational resources for the use in large-scale simulations" on page 3. Existing models are indeed complex, but this is done for a reason: for a correct biophysical description of the phenomenon and thus correct spatio-temporal dynamics of platelet activation.

6. On page 10, "simulation predicts the lag-time in distant areas, which is absent in experiment, probably indicating the presence of extra-sensitive platelets". This hypothesis is speculative, I suggest to remove "probably indicating the presence of extra-sensitive platelets" from the text, unless the Authors can substantiate it.

7. Fig.4(B,C). What variables correspond to the digits along the axes? (D,E) Same issue for the vertical axis.

Overall, I can agree with the Reviewer#2 that this paper is not suitable for PLOS Computational Biology. Clearly, it is true that "more work is needed to substantiate the conclusions of novelty and applicability in the manuscript". I also agree with the Authors that the modifications and the additional work that they have done make the manuscript suitable for PLOS ONE, as it accepts scientifically rigorous research, regardless of novelty. But the concerns mentioned above must be addressed before acceptance.

6. PLOS authors have the option to publish the peer review history of their article (what does this mean?). If published, this will include your full peer review and any attached files.

Reviewer #1: No

Reviewer #2: No

---

## [Author Response · Author response to Decision Letter 0]

17 Jul 2024

We thank the reviewers for their constructive comments. Our answers are below.

Reviewer #1:

Authors study a novel system of photo-released ADP and platelet activation in space and time. The experiments and data analysis are conducted with care.

Major comments.

The paper does not appear to include conditions where agonist induced ADP release and thromboxane synthesis play an autocrinic role. Authors are encouraged to find conditions of platelet density, P2Y12 inhibitors, or COX1 inhibitors that may modulate the spread of platelet activation. Are there conditions where platelet release of ADP causes a self-sustaining wave of calcium activation, distinct from the diffusional spread of photo-released ADP. As a thrombus grows, the platelet density is very high and autocrine activation definitely occurs.

We agree with the reviewer that autocrine (and paracrine) activation plays significant role in the thrombus growth. Moreover, our motivation and primary goal for the described experiments was to observe the self-sustaining wave of calcium activation. Unfortunately, in our experimental conditions there is no evidence for autocrine activation, and the simplest model of ADP-diffusion–mediated activation is consistent with the experimental data. We agree that it would be very interesting to find experimental conditions where the autocrine activation is directly visible, but we have not succeeded in this to date and should leave it for the future research.

The authors should discuss the role of geometry and platelet density. The height of the liquid determines the size of the reaction volume and the amount of photo-released ADP. Different results would be obtained in different geometries such as activation of a cross section of a tubular geometry or rectangular geometry (as in microfluidics).

The height of the liquid in each sample was 2 mm (we have indicated this into the Mat&Meth section). This is smaller than the laser spot waist length (Rayleigh length), which is 3.5 mm. Therefore, in these conditions the system can be considered two-dimensional, i.e. the laser intensity is independent on the vertical coordinate.

On the other hand, the width of the sample is 7 mm, which is much larger than the width of the laser spot (20 microns) and the size of the zone where the activation occurs. It means that in our conditions the boundaries have no impact on the process.

Indeed, as the Reviewer indicates, different results could be obtained in different geometries if the boundaries would be close to the activation spot, which can be true in the case of microfluidics.

Phosphate buffer will cause calcium to precipitate. The studies should be repeated with HEPES buffered saline. (Tris should be avoided since it has anti-protease activity in coagulation studies).

As the reviewer suggested, we performed additional experiments to compare the effect of two buffers, PBS and HEPES. Interestingly, no activation was observed when using HEPES buffer, whereas in the same experiment with PBS the effect was clearly visible. On the other hand, platelets looked brighter in HEPES solution. While the cause of this result is not completely clear, on the current stage we stick to using PBS as the change of buffering system would require additional experiments. The results of the comparison experiment are shown in the Supplementary materials, Figure S3. We also added a reference to these results to the Mat&Meth section.

The authors should provide evidence that the platelets have settled to the bottom of the well. If the platelets have spread then they are partially activated before the laser is used.

We agree with the Reviewer that spread platelets are partially activated prior to the experiment. Such cells are indeed visible in some experiments as a background noise. However, the majority of platelets are chaotically moving during the experiment (as can be seen in Supplementary videos), which means that they are not attached to the bottom of the well.

Minor Comments:

Abstract: “However, existing models are too complicated for

large-scale problems, for instance, simulation of the thrombus growth, where the

concentration of agonist varies significantly and heterogeneity of platelets should be

taken into account.” This statement is not correct. There are several publications that solve for thrombus growth with agonist gradients and individual platelets.

The term “existing models” in the sentence refers to the detailed biochemical models of platelet activation such as described by Purvis et al (https://doi.org/10.1182/blood-2008-05-157883 ). We corrected the text in the abstract. We also added references to several publications concerning thrombus growth simulation into the introduction.

Reviewer #2

The manuscript reports a study of platelet activation by a point source of ADP. The Authors designed an experimental setup and developed a mathematical model for spatio-temporal dynamics of the diffusion-mediated spread of the area of activated cells. The Materials and Methods are detailed enough to allow the reproducibility of the results. The Authors replied all the remarks from the previous round of peer-review, even added new experiments with healthy donors, as was required by the Reviewer #1.

1. The paper in written in standard English, however, some typos are present, e.g., on page 2, line 13 - "Biochemical cascade of the platelet activation involve special" - should be "involves".

Fixed.

2. In the PDF version of the manuscript some equations (e.g. 1 and 2) are not displayed properly, although in the docx-file the equations are readable. This should be elaborated during the publication process in case of acceptance.

This issue seems to be related to the online docx -> pdf conversion system.

3. The Authors "presumed that the displacement of the platelets as well as their size is insignificant". I can agree that this assumption is somehow valid in terms of the displacement. However, the independence of the activation threshold (minimal ADP concentration) from platelet size should be substantiated, as it is not obvious. Could there be a correlation between the ADP-threshold and platelet size? Some references suggest that platelet volume may be a risk marker for platelet activation:

https://doi.org/10.1080/095371002220148332

https://doi.org/10.1093/rheumatology/kew437

In the mentioned sentence, size was assumed to be negligible compared to the ADP change scale. We have changed the sentence as follows:

We presume that the displacement of platelets as well as their size is insignificant and their size is small compared to characteristic scale of ADP concentration change; therefore, all the objects are considered immobile and point-like.

4. Fig.3. In the caption it should be explicitly mentioned that panels E-J correspond to the areas 1-6 marked on the panel B.

Fixed.

5. The Authors claim that existing mathematical approaches are too complicated to be used in systems biology models. But in my opinion their model oversimplifies the picture. Generally, the paper tells us that the diffusion of ADP activates the platelets. The model is based on the solution of the diffusion equation (Fick's law) and the probabilistic distribution of platelet susceptibility to ADP. This finding does not reveal any new biophysics mechanism, nor it incorporates already known features into the suggested model, such as granule release and positive feedback loops. In my opinion, the paper is valuable for its experimental part (as the results seem rigorous), while the proposed model is essentially phenomenological and it doesn't enhance our understanding of platelet activation.

Therefore, I suggest to modify the title: the experimental part should brought forward, while the theoretical model is subsidiary. The fact that the theory "captures main features of experiment (activation spread) but cannot accurately describe other features" mainly says that the model is not precise enough, as it doesn't account for possible nonlinear effects and feed-back loops, which are essential for such an active medium. Clearly, the spreading of "platelet activation" in not entirely governed by the diffusion, although the diffusion of ADP indeed triggers the observed processes.

We suggest the following title:

Platelet activation near point-like source of agonist: experimental insights and computational model.

I also suggest to remove the phrase "detailed models require too much computational resources for the use in large-scale simulations" on page 3. Existing models are indeed complex, but this is done for a reason: for a correct biophysical description of the phenomenon and thus correct spatio-temporal dynamics of platelet activation.

We have removed the phrase.

6. On page 10, "simulation predicts the lag-time in distant areas, which is absent in experiment, probably indicating the presence of extra-sensitive platelets". This hypothesis is speculative, I suggest to remove "probably indicating the presence of extra-sensitive platelets" from the text, unless the Authors can substantiate it.

We have removed the phrase.

7. Fig.4(B,C). What variables correspond to the digits along the axes? (D,E) Same issue for the vertical axis.

We have added the corresponding descriptions of the axes.

Overall, I can agree with the Reviewer#2 that this paper is not suitable for PLOS Computational Biology. Clearly, it is true that "more work is needed to substantiate the conclusions of novelty and applicability in the manuscript". I also agree with the Authors that the modifications and the additional work that they have done make the manuscript suitable for PLOS ONE, as it accepts scientifically rigorous research, regardless of novelty. But the concerns mentioned above must be addressed before acceptance.

---

## [Editor Report · Decision Letter 1]

30 Jul 2024

Platelet activation near point-like source of agonist: experimental insights and computational model

PONE-D-24-12691R1

Dear Dr. Moskalensky,

We’re pleased to inform you that your manuscript has been judged scientifically suitable for publication and will be formally accepted for publication once it meets all outstanding technical requirements.

Kind regards,

Heather Faith Pidcoke, MD, MSCI, PhD

Academic Editor

PLOS ONE
---

## [Editor Report · Acceptance letter]

2 Aug 2024

PONE-D-24-12691R1 

PLOS ONE

Dear Dr. Moskalensky, 

I'm pleased to inform you that your manuscript has been deemed suitable for publication in PLOS ONE. Congratulations! Your manuscript is now being handed over to our production team.

Kind regards, 

on behalf of

Dr. Heather Faith Pidcoke 

Academic Editor

PLOS ONE